# CAN WORLD MODELS BENEFIT VLMS FOR WORLD DYNAMICS?

## ABSTRACT

Trained on internet-scale video data, world models are increasingly recognized as powerful world simulators that can generate consistent and plausible dynamics over structure, motion, and physics. While recent studies have explored the few-shot learning capabilities of world models on vision tasks, these explorations typically lack a systematic investigation of the further applicability of such methods on generic tasks. We study what happens when these priors are transferred into a Vision-Language Model (VLM): we re-purpose a video diffusion model as a *generative encoder*, queried for a single denoising step, and treat the resulting latents as an additional set of visual embeddings. We empirically investigate this class of models, which we refer to as World-Language Models (WorldLMs), and we find that generative encoders can indeed capture latents useful for downstream understanding, showing distinctions from conventional vision encoders. Naming our best-performing WorldLM **Dy**namic **V**ision **A**ligner (**DyVA**), we further discover that this method significantly enhances spatial reasoning abilities and enables single-image models to perform multi-frame reasoning. Through the curation of a suite of spatial evaluation sets, we find DyVA to surpass both open-source and proprietary baselines on out-of-domain tasks, achieving **state-of-the-art performance on MindCube**. Finally, we systematically explore extensive model designs to highlight promising directions for future work. We hope our study can pave the way for a new family of VLMs that leverage priors from world models.

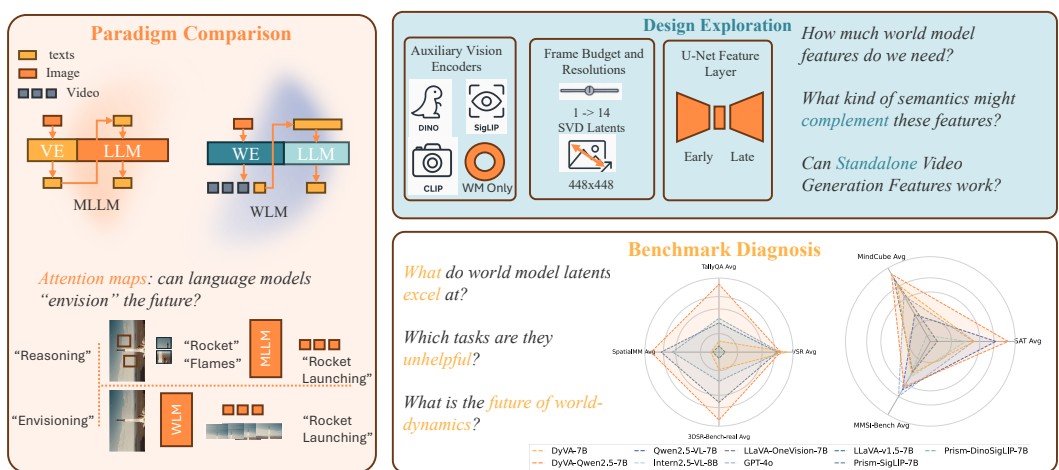

Figure 1: Our analysis is structured around three complementary lenses: (i) Paradigm comparison, contrasting static encoders (e.g., ViT/SigLIP) with world-model encoders (SVD) to ask whether VLMs can "envision" futures; (ii) Design-space exploration, probing auxiliary encoders, frame budgets, resolutions, and U-Net feature layers to understand how much and which type of world-model features matter; and (iii) Benchmark diagnosis, which reveals where world-model latents excel (e.g., spatial and multi-frame reasoning) and where they remain unhelpful (e.g., language-heavy tasks). Together, these pillars provide a discussion-oriented framework for understanding the role of world-model priors in vision–language reasoning.

## 1 INTRODUCTION

World models, originally proposed in cognitive science to explain how humans predict and plan in their environments (Tolman, 1948), have recently emerged as powerful tools in machine learning. Generative world models (Agarwal et al., 2025b; OpenAI, 2024; Wan et al., 2025; Hu et al., 2023; Blattmann et al., 2023; Yang et al., 2025b; Guo* et al., 2023; 2025; Chen* et al., 2025), such as video generation models (VGMs), trained on internet-scale data, encode strong priors over objects, spatial layouts, and dynamics. These priors allow them to predict plausible future scenarios that are consistent in 3D structure and physically coherent in motion

However, a largely overlooked implication of World Models is that the ability to generate coherent futures signals a form of semantic understanding of visual dynamics; this difference between visual generation and understanding has shaped a decade of representation learning. This suggests that world models can be more than generators—they may serve as transferable encoders that enrich downstream tasks with spatial, temporal, and predictive signals. As a result, recent works have attempted to use video generation backbones for visual perception tasks (Acuaviva et al., 2025).

In this work, we ask a fundamental question: ***To truly understand the world, must a model first learn to predict?***

To empirically investigate this, we introduce a simple yet effective framework on Vision–Language Models (VLMs). We specifically explore by evaluating the applicability of predictive world models on a generic task—Visual Question Answering (VQA)—to assess their broader potential as **generalizable vision encoders**. Currently, mainstream VLMs primarily rely on ViT-based encoders such as CLIP (Radford et al., 2021), SigLIP (Zhai et al., 2023), and DINO (Caron et al., 2021; Oquab et al., 2024), which extract visual semantics from image patches and are then projected as visual tokens into language backbones. While these encoders are semantically aligned, they are limited by temporal reasoning and weaken spatial grounding when multiple views or sequential cues are present. On the other hand, we re-purpose a world model (Stable Video Diffusion SVD) as a novel **Generative Encoder**. Our core mechanism is to extract latent features from a **single denoising step** of its U-Net. This single step, we hypothesize, captures the low-dimensional world-dynamics prior sufficient for downstream understanding. These dynamics-aware latents are then fused with static image features (e.g., SigLIP) and projected into the LLM. The design is very efficient: all encoders remain frozen, with only lightweight projectors and the LLM being trained.

To this end, we conduct a systematic investigation comparatively evaluating this class of models, which we refer to as World-Language Models (WorldLMs). Our findings are as follows:

- **Shift in Reasoning Paradigm.** The generative prior alters the model's reasoning process. It moves beyond describing static content to envisioning dynamic possibilities.
- **Zero-shot Multi-Frame Adaptation.** Trained exclusively on single images, the generative encoder enables emergent multi-frame reasoning without multi-image pre-training.
- **State-of-the-Art Zero-Shot Reasoning.** On multi-frame benchmarks, DyVA achieves state-of-the-art performance, decisively outperforming leading proprietary models such as Qwen2.5-VL (Bai et al., 2025) and GPT-4o (OpenAI et al., 2024).

Our best-performing WorldLM variant, **Dynamic Vision Aligner** (**DyVA**), exemplifies this paradigm shift. In zero-shot evaluations on challenging multi-frame reasoning benchmarks, DyVA decisively surpasses even proprietary models, for instance, a **28.3%** lead on the **MindCube** benchmark over the GPT-4o model. This provides strong evidence that the ability to predict is a powerful, perhaps essential, foundation for stronger representation learning.

As shown in Figure 1, we systemically organize our investigation revolving around three pillars:

**Paradigm comparison.** World-model encoders versus static encoders reveal distinct strengths: world-model latents benefit spatial and multi-frame reasoning, while static encoders excel on semantics-heavy benchmarks.

**Benchmark diagnostics.** Through curated evaluation sets—including MindCube Yin et al. (2025), SAT-Bench Ray et al. (2024), VSR Liu et al. (2023a), we find DyVA to surpass both open-source and proprietary baselines on out-of-domain tasks, achieving **state-of-the-art performance on Mind-Cube and SATBench**. —we show that dynamics-aware latents particularly help with object rela-

tions, perspectives, and multi-frame spatial reasoning, while offering less gain on tasks requiring stronger language priors.

**Design-space exploration.** We analyze different encoder setups to identify when predicted latents help or hinder performance, laying the groundwork for a new class of WorldLMs exploiting world-model priors.

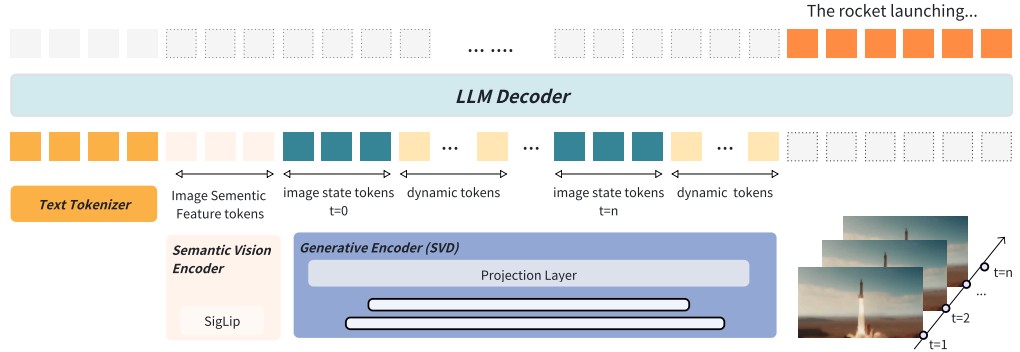

Figure 2: **WorldLM Pipeline.** A SigLIP encoder extracts static semantic features from the input image. Concurrently, a Generative Encoder generates dynamic state tokens and predicts future dynamic tokens to capture temporal changes, using evenly spaced keyframe slots. All visual tokens are projected into a shared embedding space, concatenated with text tokens, and then fed into the LLM decoder.

## 2 PRELIMINARY

To ground our analysis, we need 1) a framework to incorporate the dynamic features of a world model into a multimodal language model (which we term **WorldLM**), 2) a training recipe, and 3) an implementation of inference supporting both single- and multi-image datasets.

**Framework.** Traditional VLMs like LLaVA (Liu et al., 2024), QwenVL (Bai et al., 2025), and Prismatic-VLMs (Karamcheti et al., 2024), adopt an architecture consisting of three core components. Given an input image $x_{img} \in \mathbb{R}^{H \times W \times C}$ and a text prompt $u_{prompt}$, the model processes them through the following components:

- **Semantic Vision Encoder.** The input image $x_{img}$ is processed by a frozen pre-trained ViT-based (Dosovitskiy et al., 2021) vision encoder $V_\omega$, for example SigLIP (Zhai et al., 2023), to extract a sequence of feature embeddings $p_{img} = V_\omega(x_{img})$, where $p_{img} \in \mathbb{R}^{L \times d_{vision}}$.

- **Projector.** The visual features $p_{img}$ are subsequently mapped into the language model's embedding space by a projector $F_\psi$. This yields a sequence of embeddings $e_{img} = F_\psi(p_{img})$, where $e_{img} \in \mathbb{R}^{L \times d_{text}}$. The projector is typically implemented as a simple Multi-Layer Perceptron (MLP) with GELU activations (Hendrycks & Gimpel, 2023).

- **LLM Backbone.** Finally, the language model $LM_\theta$ autoregressively generates the textual output $u_{out}$. It is conditioned on the concatenated sequence of the projected image features $e_{img}$ and the text prompt embeddings $e_{prompt}$: $u_{out} = LM_\theta([e_{img}; e_{prompt}])$

To obtain the dynamic visual information and motion priors of the input image, we employ another component to encode it:

- **Generative Encoder.** We utilize Stable Video Diffusion (SVD) (Blattmann et al., 2023) as our encoder. SVD consists of a VAE (Kingma & Welling, 2022) encoder $\phi$ and a U-Net (Ronneberger et al., 2015) denoiser $f_\theta$. The input image $x_{img}$ is first encoded by VAE into a latent $z_0$, which is then replicated $T$ times to form an initial video latent $Z_0$. A single Euler integration step is then applied to yield an updated latent $Z_1 = Z_0 + \Delta\sigma\, f_\theta(Z_0, \sigma_0, c)$.

Rather than rendering video frames, the final output $D_{img} = \text{Hidden}^{\text{pre-mid}}(f_\theta, Z_1)$ is extracted from the U-Net's pre-middle block.

As is shown in Fig. 2, semantic features $p_{img}$ and dynamic features $\tilde{H}$ are projected by two separate projectors $P_{\text{sem}}$ and $P_{\text{dyn}}$ into the LLM space, yielding $V_s = P_{\text{sem}}(p_{img}) \in \mathbb{R}^{L_s \times d}$ and $V_d = P_{\text{dyn}}(\tilde{H}) \in \mathbb{R}^{L_d \times d}$. The fused sequence is $V = [V_s; V_d] \in \mathbb{R}^{(L_s + L_d) \times d}$, which, together with prompt embeddings $E_{prompt}$, is fed into the LLM backbone to autoregressively generate answer tokens $u_{out} = \text{LM}_\theta([V; E_{prompt}])$. By fusing both streams, our WorldLM leverages static semantics (from SigLIP) and dynamics-aware priors (from SVD) for multimodal reasoning.

**Training recipe.** We adopt the training strategy from Prismatic-VLMs (Karamcheti et al., 2024) using a single-stage training to align modalities and incorporate generative latents: We jointly train both the projectors and the LLM on a mixture of multimodal instruction datasets from LLaVA-1.5 (Liu et al., 2023b), together with examples from established vision-language benchmarks (e.g., GQA (Hudson & Manning, 2019), TextCaps (Sidorov et al., 2020)), and language-only samples from ShareGPT (sha). This training paradigm not only effectively aligns the generative encoder's representations with the semantic space of the language backbone but also improves the model's compositional generalization, enabling it to reason over both motion priors and static features. Remarkably, the entire training process completes in only 10.3 hours on $16 \times$A800 GPUs ($\approx$165 GPU-hours) while achieving competitive performance, underscoring the efficiency of our approach.

**Inference Protocol** During inference, we employ SigLIP-so400m-patch14-224 as the semantic vision encoder and Stable Video Diffusion as the generative encoder, with the image resolution set to $448 \times 448$. As shown in Fig. 2, or $K$ input images, we allocate key frames using evenly spaced indices within the $T$-frame latent tensor, replace the corresponding slots with encoded keyframes before the Euler step, and reuse the resulting latents as additional visual tokens. For the semantic vision encoder, only the first input image is encoded and concatenated with the input of the generative encoder. Unless otherwise specified, the number of frames ($T$) is set to 8 for both single-image and multi-image inputs.

Following the proposed framework, training setup, and inference principles, we train a family of WorldLM models and designate the ones excelling in **Dy**namic **V**ision **A**lignment as *DyVA*.

## 3 PARADIGM COMPARISON

**Do WorldLM Encoders Entail Visual Semantics Understanding?**

In this section, we explore how world model latents can benefit visual understanding by contrasting two differentiating encoder paradigms: (i) traditional static encoders such as CLIP and SigLIP that prioritize multi-modal semantic alignment, and (ii) world-model encoders based on video generation models that generate dynamics-aware latents. We begin by comparing the most intuitive design to test if WorldLMs can work, by directly replacing the CLIP vision encoder of LLaVA 1.5 (Liu et al., 2024) with a Generative Encoder(e.g., SVD) following the WorldLM pipeline settings in Fig. 2.

**Generative encoders exhibit fundamentally different performance.**

We begin with a motivating case study, as illustrated in Fig. 3. Models leveraging static encoders, such as LLaVA, adopt a *reasoning* paradigm. The output of LLaVA tends to be more descriptive, describing in depth the details of the given image input. WorldLM, on the other hand, employs an *envisioning* paradigm, not only encodes the current state of the image, but it also performs a prediction of plausible (e.g., "will drive away", and "drive to the other rover"). This case reveals an inherent difference between the two paradigms: This case reveals an intrinsic difference between the two paradigms: VLM reasons by the given image's embeddings, whereas WorldLM attends to depict the embeddings of generated predictions.

**Multi-frame capture more useful semantic features than Single Frame.** The quantitative 3 comparison between using different numbers of generated dynamic latents shows its effect on downstream tasks. When the generated frames of the video prediction model increase from 1 to 14, we see a general increasing trend.

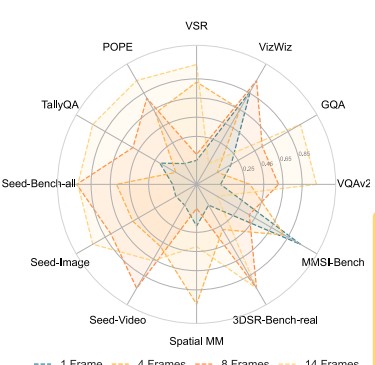
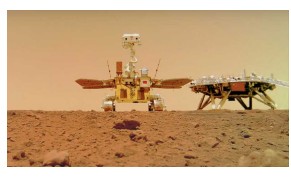

Figure 3: **Paradigm Comparison.** Using the most straightforward setup of WorldLM, we evaluate the impact of predicting 1, 4, 8, and 14 frames. The radar chart demonstrates that increasing the number of frames consistently boosts performance across various vision-language tasks, especially in spatial and temporal reasoning benchmarks such as SeedBench, VSR, and TallyQA. The qualitative example further illustrates that under the same configuration, our WorldLM exhibits a distinct reasoning paradigm by envisioning—offering more concise descriptions, stronger spatial grounding, and more structured temporal foresight compared to LLaVA's reasoning methods.

**Meanwhile, it performs great on spatial-reasoning tasks.** Notably, the gains are most pronounced on benchmarks demanding sophisticated spatio-temporal reasoning, such as *SeedBench*, *VSR*, and *TallyQA*. WorldLM's generative encoders do entail visual understanding, especially in spatio-temporal reasoning. This demonstrates the potential of using world models as dynamics-aware encoders to allow VLMs with a deeper, more grounded level of spatial understanding.

**Limitations of WorldLMs.** Despite the clear advantages in temporal reasoning, our analysis reveals a critical trade-off. The case study in Figure 3 offers a qualitative explanation for this phenomenon. While our world model correctly grounds the spatial structure of the scene (e.g., "rocket on the ground... large rocket in the distance"), it hallucinates the semantic identity of the objects, misidentifying the Mars lander and rover as "rockets". Therefore, we believe that using a world model as an encoder has the potential to enhance predictive and spatial reasoning tasks, but requires further improvement to ensure basic semantic capabilities.

## 4 BENCHMARK ANALYSIS: INVESTIGATION

### 4.1 EXPERIMENTAL SETUP

We document the configurations, datasets, and training protocols underlying our study. Unless otherwise noted, all settings use a 7B-parameter LLaMA-2 LLM backbone, with both SigLIP and SVD encoders frozen during a single-stage instruction tuning. Training updates are restricted to lightweight projection layers and the language backbone.

### 4.2 DATASETS AND EVALUATION TARGETS

Benchmarks vary widely in their emphasis on *spatial grounding*, *temporal coherence*, and *semantic understanding*. To assess these dimensions, we curate a suite of open-source **out-of-domain (OOD)** datasets on which our models have not been trained. This allows us to isolate the transferability of world-model priors.

**Single-image spatial reasoning.** We evaluate on benchmarks that probe relational and spatial understanding without temporal context, including VSR (Liu et al., 2023a), TallyQA (Acharya et al., 2018), SpatialMM-Obj (Shiri et al., 2024), and 3DSR-Bench-real (Ma et al., 2025). Baselines include LLaVA-1.5 (Liu et al., 2024), Prism-SigLIP-7B (Karamcheti et al., 2024), and Prism-DinoSigLIP-7B (Karamcheti et al., 2024).

**Multi-image and temporal reasoning.** To assess robustness to sequential inputs and temporal structure, we use MMSI-Bench (Yang et al., 2025a), SAT-Synthetic (Ray et al., 2024), and Mind-Cube (Yin et al., 2025). These benchmarks require models to integrate cues across frames or viewpoints, testing whether world-model latents can enable multi-frame reasoning. We compare against both open-source and proprietary large-scale VLMs, including Qwen-2.5-VL-7B (Bai et al., 2025), InternVL-2.5-7B (Chen et al., 2025), LLaVA-OneVision-7B (Li et al., 2024), and GPT-4o (OpenAI et al., 2024). Note that all of the compared benchmarks are trained with multi-frame or video data, whereas we train on single images only.

## 4.3 Experimental Analysis and Insights

Table 1: Performance comparison between DyVA and state-of-the-art methods on multi-image benchmarks SAT Synthetic, MMSI-Bench, and MindCube. DyVA outperforms baselines in these Out-of-Domain tasks. The highest average values are in bold.

| Model | SAT Synthetic | | | | | | MindCube | | | |
|---|---|---|---|---|---|---|---|---|---|---|
| | Obj Move. | Act. Seq. | Act. Cons. | Goal Aim | Persp. | Avg. | Rot. | Among | Around | Avg. |
| Qwen2.5-VL-7B | 79.29 | 84.70 | 47.83 | 25.84 | 35.17 | 53.16 | 38.76 | 29.50 | 21.35 | 29.26 |
| Intern2.5-VL-8B | 77.74 | 55.49 | 53.74 | 15.03 | 32.61 | 48.06 | 18.68 | 36.45 | 18.20 | 18.68 |
| LLaVA-OneVision-7B | 71.10 | 21.64 | 49.85 | 31.76 | 35.43 | 43.24 | 36.45 | 48.42 | 44.09 | 47.43 |
| GPT-4o | 61.50 | 33.20 | 47.60 | 67.50 | 37.50 | 49.40 | 40.17 | 29.16 | 38.81 | 38.81 |
| **DyVA-7B** | 49.15 | 57.81 | 49.25 | 53.38 | 40.44 | 49.51 | 37.70 | 43.10 | 49.00 | 44.62 |
| **DyVA-Qwen2.5-7B** | 78.83 | 62.13 | 49.85 | 51.86 | 41.72 | **55.24** | 37.20 | 39.10 | 51.70 | **49.80** |

| Model | MMSI-Bench | | | | | | | | | | | |
|---|---|---|---|---|---|---|---|---|---|---|---|---|
| | Positional Relationship | | | | | | Attribute | | Motion | | MSR | Avg. |
| | Cam–Cam | Obj–Obj | Reg–Reg | Cam–Obj | Obj–Reg | Cam–Reg | Means | Appr | Cam | Obj | | |
| Qwen2.5-VL-7B | 32.3 | 27.7 | 29.6 | 32.6 | 24.7 | 32.5 | 26.6 | 27.3 | 16.2 | 31.6 | 30.3 | 28.70 |
| Intern2.5-VL-8B | 24.7 | 24.5 | 24.7 | 25.6 | 29.4 | 26.5 | 25.0 | 18.2 | 20.3 | 39.5 | 25.8 | 25.90 |
| LLaVA-OneVision-7B | 20.4 | 33.0 | 29.6 | 29.1 | 25.9 | 30.1 | 29.7 | 25.8 | 18.9 | 34.2 | 11.6 | 24.50 |
| GPT-4o | 34.4 | 24.5 | 23.5 | 19.8 | 37.6 | 27.7 | 32.8 | 31.8 | 35.1 | 36.8 | 30.8 | **30.30** |
| **DyVA-7B** | 21.5 | 30.9 | 25.9 | 31.4 | 27.1 | 20.5 | 35.9 | 24.2 | 13.5 | 19.7 | 24.2 | 24.90 |
| **DyVA-Qwen2.5-7B** | 15.1 | 33.0 | 25.9 | 33.7 | 35.3 | 30.1 | 32.8 | 25.8 | 17.6 | 27.6 | 29.3 | **28.00** |

Tab. 1 and 2 present representative results under both single- and multi-image settings. This framing allows us to disentangle how world-model features contribute across different reasoning regimes.

As presented in Tab. 1 and 2, we evaluate the OOD performance of DyVA-LLaMA-7B and DyVA-Qwen-2.5-7B. We examine DyVA's performance relative to existing vision-language models across various spatial reasoning tasks. The key differences lie in DyVA's use of "world-model latents" (SVD-based latent tokens fused with SigLIP image features) versus baselines that use only standard visual embeddings. Below, we discuss the strengths and weaknesses of DyVA in each benchmark category, drawing on the reported results and known properties of these tasks and models.

The key findings are summarized as follows:

**DyVA can enable single-image trained WorldLMs to perform multi-image tasks exceptionally well.** As in Tab. 1, our best variant can perform strongly in multi-frame spatial understanding tasks.

Specifically, on the MindCube benchmark (Tab. 1), DyVA-Qwen2.5 achieves a new state-of-the-art performance with the highest overall score (49.8% vs. 47.4% for the best baseline). It particularly excels in "Around" (rotating viewpoint) tasks (51.7% vs. 44.1%) and matches or slightly exceeds baselines on "Rot" tasks (37% vs. 36%). This result suggests that DyVA latents significantly aid in tasks requiring mental rotation and perspective-taking, likely because they encode cross-view consistency—the world model inherently "knows" how an object appears from different angles.

This achievement is especially noteworthy considering the training efficiency. Compared to baselines, LlaVA-One-Vision is trained on 4M multiframe images. Intern 2.5-VL is pretrained with 16.3M samples, including multi-image and video data. Qwen-2.5-VL is also pre-trained with a variety of data comprising videos and multi-images. These baselines also have several complex methods for image preprocessing, such as patchifying (Li et al., 2024), processing at different fps (Bai et al., 2025), and high-res processing (Chen et al., 2025). In stark contrast, we trained our DyVA model using only the most basic processing methods and a minimalistic data mixture.

Table 2: Performance comparison of DyVA variants against baselines on various single-image spatial reasoning benchmarks, including VSR, TallyQA, SpatialMM-Obj, and 3DSR-Bench-real. These are Out-of-Domain tasks where models are not trained and perform zero-shot inference. Our results surpass all baseline models, indicating an improved spatial-reasoning capability from world-model predicted dynamics. Highest values are highlighted in bold.

| Models | Data | VSR | | | | | | | |
| --- | --- | --- | --- | --- | --- | --- | --- | --- | --- |
| | | Topo. | Prox. | Proj. | Direc. | Adj. | Orien. | Unall. | Avg. |
| LLaVA-v1.5-7B | 558k+665k | 52.24 | 50.00 | 54.77 | 50.00 | 50.86 | 48.98 | 57.50 | 52.94 |
| Prism-SigLIP-7B | 665k | 67.48 | 62.50 | 65.63 | 66.67 | 55.17 | 55.10 | 67.50 | 64.97 |
| Prism-DinoSigLIP-7B | 665k | 71.34 | 59.38 | 65.63 | 64.29 | 53.45 | 48.98 | 52.50 | 65.46 |
| **DyVA-7B** | 665k | 68.90 | 68.75 | 66.74 | 66.67 | 66.38 | 61.22 | 57.50 | **67.10** |
| **DyVA-Qwen2.5-7B** | 665k | 66.67 | 71.88 | 68.74 | 61.90 | 62.93 | 40.82 | 55.00 | 65.63 |

| Models | TallyQA | SpatialMM-Obj | | | 3DSR-Bench-real | | | | |
| --- | --- | --- | --- | --- | --- | --- | --- | --- | --- |
| | Avg. | 1-obj | 2-obj | Avg. | H. | L. | O. | M. | Avg. |
| LLaVA-v1.5-7B | 58.74 | 57.37 | 44.87 | 48.91 | 55.42 | 57.82 | 26.09 | 39.42 | 45.02 |
| Prism-SigLIP-7B | 62.25 | 62.54 | 46.77 | 51.86 | 52.28 | 60.22 | 27.23 | 42.17 | 46.55 |
| Prism-DinoSigLIP-7B | 62.93 | 58.56 | 47.72 | 51.22 | 56.85 | 59.42 | 27.23 | 38.97 | 45.82 |
| **DyVA-7B** | 59.47 | 54.78 | 46.29 | 49.03 | 53.71 | 57.60 | 27.23 | 40.80 | 45.41 |
| **DyVA-Qwen2.5-7B** | **68.11** | 62.74 | 47.53 | **52.44** | 52.57 | 54.51 | 27.23 | 49.60 | **47.16** |

Our modest training budget and intuitive multi-image inference method suggest that world model latents strongly enhance the spatial understanding on multi-image benchmarks. We also believe that the fusion of SVD with SigLIP is a key factor that directly improves multi-image reasoning abilities.

**DyVA excels in handling spatial relations, counting and object queries, and 3D Scenes.** In Single-Image Spatial Reasoning, DyVA's world-model features boost performance on tasks emphasizing geometric and relational spatial reasoning (orientation, adjacency, multi-object spatial layouts), reflecting improved 3D awareness.

1. Visual Spatial Relations (VSR): DyVA-LLaMA (SigLIP+SVD) achieves the highest average score (67.1%) across VSR subtasks (topology, proximity, projection, direction, adjacency, orientation, unaligned), outperforming the SigLIP-only baselines (64.9–65.5%) Tab. 2. In particular, DyVA significantly improves orientation reasoning (61.2% vs 55–49% for baselines) and proximity/topology, suggesting world-model latents better encode spatial layouts and object alignment. However, DyVA falls behind on the "Unaligned" subtask (57.5% vs 67.5% for Prism-SigLIP), indicating that embedding world-model context can hurt when objects lack canonical alignments (perhaps because the latent prior biases toward canonical scene structures).

2. Counting and Object Queries (TallyQA, SpatialMM-Obj): On TallyQA (visual counting), DyVA-Qwen2.5 excels (68.1% average), well above Prism baselines (62–63%) and LLaVA (58.7%)Tab. 2. This suggests the latents help Qwen2.5 better aggregate multi-object cues needed for counting. Interestingly, DyVA-LLaMA does not show the same gain (59.5%), implying that effective use of SVD features depends on backbone capacity. For the SpatialMM-Obj task (single- vs multi-object queries), DyVA-Qwen2.5 again slightly outperforms others (52.4% vs 51.8% baseline) on the combined 1- and 2-object questions.

3. 3D Scene Reasoning (3DSR-Bench-real): This benchmark measures 3D spatial and depth understanding in real images. Notably, DyVA greatly improves the "Multiple objects" (M) subset (49.6% vs 40% for baselines). This aligns with the idea that SVD latents capture implicit depth and occlusion cues learned from video/world modeling

**Limitations and Areas for Improvement.** Despite its strengths in spatial reasoning, DyVA exhibits certain limitations, particularly on tasks that rely heavily on semantic language priors, non-canonical object arrangements, or temporal sequence understanding.

1. Weakened Performance on Language-Intensive Tasks: The fusion of world-model tokens can dilute the semantic precision required for certain tasks. On benchmarks like VQAv2 and TextVQA,

which demand strong language priors and OCR capabilities, DyVA underperforms compared to SigLIP-only baselines. This suggests that while SVD latents enhance spatial awareness, they can interfere with fine-grained semantic grounding and text recognition where the original visual features are more direct and precise.

2. Bias Towards Canonical Scene Structures: As previously noted in the VSR analysis, DyVA's performance drops significantly on the "Unaligned" subtask (57.5% vs. 67.5%). This indicates that embedding world-model context can be detrimental when objects lack canonical alignments. The model's latent prior appears biased toward common or expected scene structures, hindering its ability to reason about novel or unusual spatial configurations.

3. Less Reliable Sequential and Temporal Reasoning: The current SVD latents are less effective for understanding dynamic sequences. This is evidenced by a large performance drop in SAT Action Sequence and mixed results on MMSI. These outcomes suggest that the latents, while powerful for static scenes, are less reliable for predicting discrete action orders or interpreting rapid changes over time, marking a clear area for future improvement.

## 5 DESIGN-SPACE EXPLORATION: WHY DYVA WORKS?

Building on the strong spatial performance demonstrated in both single-image and multi-image tasks in our experiments, we further analyze two key design axes to investigate the sources of WorldLM's benefits: (i) the choice of different semantic vision encoders, and (ii) the potential of leveraging text-loss to jointly supervise the training of the VAE and U-Net.

Table 3: **Performance Comparison of SVD-based Vision Models.** Benchmark scores across a suite of VQA, reasoning, and spatio-temporal tasks. All experiments use the LLaMA-2 7B backbone. The highest score in each column is marked in **bold**, and the second-highest is underlined. Align: one-time alignment on LAION-558k. F1: one-time finetuning. Fused: 3-layer MLP projector.

| Model | Align | VQAv2 | GQA | VizWiz | VSR | POPE | TallyQA | SeedBench | SpatialMM | 3DSR |
|---|---|---|---|---|---|---|---|---|---|---|
| **VAE-Only** | × | 46.98 | 40.53 | 38.90 | 52.04 | 66.42 | 39.55 | 38.18 | 38.81 | 44.15 |
|  | ✓ | 50.70 | 43.26 | 48.67 | 52.29 | 60.80 | 42.48 | 41.53 | 37.3 | 43.43 |
| **SVD-Only** | × | 63.51 | 55.18 | 44.95 | 57.93 | 82.38 | 49.75 | 50.15 | 42.03 | 42.93 |
|  | ✓ | 61.82 | 50.20 | 50.60 | 53.60 | 75.61 | 53.27 | 52.55 | 40.60 | 43.50 |
|    U-Net Trainable | ✓ | 63.36 | 54.49 | 50.24 | 57.93 | 79.88 | 51.51 | 52.76 | 40.80 | 43.43 |
|    U-Net & VAE Trainable | ✓ | 60.99 | 49.80 | 50.17 | 52.53 | 77.08 | 53.75 | 52.33 | 39.50 | 44.00 |
| **Dino + SVD** | × | 68.77 | 58.50 | 50.73 | 62.52 | 85.25 | 52.78 | 55.19 | 44.79 | 44.26 |
|  | ✓ | 68.44 | 55.57 | 51.13 | 59.41 | 85.54 | 54.15 | 56.49 | 43.40 | 45.07 |
| **SigLIP + SVD** | × | **75.36** | **61.52** | **55.95** | **67.10** | 85.97 | **59.47** | **66.61** | **49.03** | 45.40 |
|  | ✓ | 73.63 | 58.89 | 54.63 | 61.62 | 84.37 | 56.98 | 62.09 | 45.40 | 45.49 |
|    U-Net Trainable | ✓ | 74.02 | 59.86 | 54.60 | 62.27 | 85.61 | 57.42 | 63.39 | 45.95 | 44.11 |
| **CLIP + SVD** | × | 73.51 | 59.67 | 53.14 | 64.89 | 85.80 | 58.25 | 65.45 | 46.07 | **46.13** |
|  | ✓ | 72.99 | 60.74 | 55.89 | 65.38 | 85.80 | 55.37 | 65.33 | 46.70 | 44.42 |
| **DinoSigLIP + SVD** | × | 74.28 | 60.16 | 54.13 | 64.81 | **87.27** | 57.42 | 64.54 | 48.65 | 44.15 |
|  | ✓ | 72.42 | 59.28 | 54.47 | 61.29 | 86.75 | 54.98 | 61.54 | 47.00 | 45.14 |

### 5.1 WHY DO VAE, DINO, SVD-ONLY NOT WORK, BUT SIGLIP+SVD DOES?

To investigate the respective roles of the generative encoder and the semantic vision encoder within WorldLM, we conducted a two-stage ablation study: **First**, in a setting without the semantic vision encoder, we decoupled the generative encoder into its constituent VAE and the complete generative encoder architecture. We then trained and comparatively evaluated the performance of two distinct encoding approaches: one employing only the VAE for encoding and the other utilizing the entire generative encoder (SVD). **Second**, while keeping the generative encoder fixed, we systematically substituted the backbone of the semantic vision encoder with various alternative architectures to analyze its impact on the model's overall performance.

Our quantitative experimental results are presented in Tab. 3. Furthermore, to provide a more intuitive visualization and comparison of the performance of different encoders.

**Prediction Matters.** The inference protocol for the SVD encoder has been detailed in Sec. 2. A similar inference process is employed when using the VAE as the generative encoder. In contrast to

extracting features from the pre-middle block of the U-Net, we directly use the features encoded by the VAE. To align the feature dimensionality with that of the SVD, we prepend several convolutional layers to the projector. As evidenced by our experimental results in Tab. 3, the model employing only VAE for encoding exhibits a performance degradation across nearly all benchmarks when compared to models using SVD. This finding underscores the significance of the predicted dynamics for the WorldLM.

**WorldLM needs a text-aligned encoder.** Although SigLIP (Zhai et al., 2023) has recently shown dominant performance as an emerging vision encoder in current state-of-the-art VLMs, such as LLaVA-One-Vision (Li et al., 2024) and Prismatic-VLM (Karamcheti et al., 2024), in this study, we investigate the respective roles of SigLIP, CLIP (Radford et al., 2021), DINOv2 (Oquab et al., 2024), and a combined DINO-SigLIP architecture as the semantic vision encoder. To ensure a fair comparison, we selected the ViT-L version for each model, all configured for a $224 \times 224$ input resolution. Furthermore, we adopted a consistent image processing strategy, which involves scaling and then cropping all images to these uniform resolutions.

As demonstrated in Tab. 3, models that utilize SigLIP (including the DINO-SigLIP combination) or CLIP as the semantic vision encoder significantly outperform the model using DINOv2. Furthermore, when considering the aforementioned investigation of the generative encoder, the model with DINOv2 as the semantic vision encoder, in turn, shows better performance than the generative-encoder-only architecture.

This leads to a key insight: for our WorldLM framework that is trained under the text-loss supervision, in addition to the predicted dynamic features, it requires supplementary visual-semantic information from a model pre-trained on language-vision tasks (i.e., a text-aligned model). This insight also paves the way for future explorations: Can the generative encoder alone suffice to replace the semantic vision encoder? Could the performance be further improved by replacing the VAE? And is text-loss supervision the answer to WorldLM training?

### 5.2 Can DyVA Benefit from U-Net & VAE Training on Text-loss?

We investigated the efficacy of fine-tuning the SVD's core components (U-Net and VAE) using only a text-loss signal. Our experimental results indicate this strategy is largely ineffective.

**Text supervision failed to help VQA tasks.** As shown in Tab. 3, making only the U-Net trainable yields inconsistent and marginal performance changes, while allowing both the U-Net and VAE to be trainable leads to a distinct and widespread degradation in performance across the benchmarks.

This suggests the high-level semantic supervision from the text-loss is ill-suited for adapting the low-level generative priors of these components. This constitutes one of the limitations of our current work. An alternative approach, inspired by methods like RAPE-E (Leng et al., 2025), involves aligning the features from the VAE and U-Net with the visual features from a semantic encoder such as DINOv2. Exploring such an alignment strategy is a promising direction for future research.

## 6 Discussions and Outlooks

**(1)** Paradigm comparisons reveal that world-model latents are powerful: generated frame latents unlock spatial and multi-view reasoning, yet can erode fidelity and increase hallucination. **(2)** Design-space sweeps clarify which architectural choices mitigate these effects, while benchmark diagnostics explain when each paradigm wins. **(3)** Open directions include aligning SVD tokens with textual references, adaptive frame selection, and attention regularizers that maintain semantic grounding while exploiting the structure of the world model. **(4)** World Model is a powerful visual encoder: The key concept of WorldLM is using the prediction pretraining to enhance the spatial-temporal understanding ability of the general VLM. Though DyVA achieves SOTA performance, one significant weakness still remains: its prediction feature relies on the worst-performing encoder, VAE. Therefore, the world model can be more deeply explored, instead of using SVD, design a world model closer to the language latent space, for understanding tasks. For example, use SigLIP to train a SigLIP world model by Joint-Embedding-Prediction-Architecture. Furthermore, we argue that using a prediction model as an encoder might be a potential new paradigm for more broader domain, across robotics, visual-language understanding, and to more general visual understanding.

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

## A    APPENDIX

## B    THE USE OF LLMS

Large language models (LLMs) were used to refine and polish writing.

## C    RELATED WORK

### C.1    WORLD MODELS

Various methods have been developed to learn predictive models of visual dynamics. Ha and Schmidhuber (2018) proposed the original World Models framework, which learns a compressed latent representation of an environment's dynamics using generative RNNs (Ha & Schmidhuber, 2018). Hafner introduced PlaNet (Hafner et al., 2018) and later Dreamer (Hafner et al., 2019), which use latent space dynamics models trained on pixel observations for planning and control. More recently, large-scale self-supervised video models have emerged. For example, Meta's V-JEPA 2 (Assran et al., 2025) and NVIDIA's COSMOS (Agarwal et al., 2025a) provide video foundation models that enable understanding, prediction, and planning from raw visual data. Zhou (2024) introduced DINO-WM, a world model that leverages pretrained DINOv2 patch features to enable zero-shot goal-reaching via planning in feature space (Zhou et al., 2024). Similarly, Stability AI's Stable Video Diffusion trains a high-capacity latent video diffusion model on vast video datasets for high-quality text-to-video and image-to-video generation (Blattmann et al., 2023).

### C.2    GENERALIST MODELS

Recent work has explored using diffusion-based generative models for flexible multi-task and in-context learning. Wang (2023) presented Prompt Diffusion, a method that enables in-context learning in diffusion models by conditioning on example input-output image pairs and a text prompt (Wang et al., 2023). Geng (2023) proposed InstructDiffusion, a unified framework that casts diverse vision tasks as a pixel-space image manipulation guided by human instructions, learned via a diffusion process (Geng et al., 2023). Bai (2024) introduced a sequential modeling approach that represents images and annotations as "visual sentences," enabling training a single large vision model across many tasks without using any language data (Bai et al., 2024). Lin (2025) presented RealGeneral, which reformulates image generation as conditional frame prediction analogous to LLM in-context learning: using video diffusion models with novel modules, they unify multiple image-generation tasks (e.g. custom generation, canny-to-image) within one framework (Lin et al., 2025).

### C.3    VISION ENCODERS FOR MLLMS

The choice of vision encoder is critical for multimodal LLMs. Radford (2021) introduced CLIP, which learns joint image–text embeddings via contrastive pretraining on large image-caption datasets (Radford et al., 2021). Building on CLIP, Tschannen (2025) developed SigLIP 2, which augments the original sigmoid-contrastive objective with self-distillation, masked prediction, and multilingual pretraining to improve semantic understanding, localization, and dense features (Tschannen et al., 2025). Caron (2021) showed that self-supervised ViT models (DINO) learn rich spatial features: their DINO model (trained with a self-distillation loss) achieves strong representation quality with emergent object-centric properties (Caron et al., 2021). Jiang (2023) evaluated various image encoders in multimodal LLMs and proposed COMM, a simple feature-merging strategy that fuses multi-layer CLIP and DINO features, demonstrating improved grounding and fine-grained visual understanding in downstream tasks (Jiang et al., 2023).

## D    APPENDIX: MODEL FORMALIZATION

**VLM basics.** A frozen SigLIP image encoder $E_{\text{siglip}}$ maps an image $x \in \mathbb{R}^{H \times W \times 3}$ to a grid of patch embeddings $S \in \mathbb{R}^{N \times C_s}$, where $N$ is the number of patches and $C_s$ the channel width. A

lightweight projector $P_{\mathrm{siglip}} : \mathbb{R}^{C_s} \to \mathbb{R}^d$ aligns these to the LLM token space:

$$V_s = P_{\mathrm{siglip}}(S) = \mathrm{MLP}_s(S) \in \mathbb{R}^{N \times d}, \tag{1}$$

where $\mathrm{MLP}_s$ is a 2-layer MLP with GELU activations.

**SVD for single-image → video.** Stable Video Diffusion (SVD) consists of a VAE encoder $\phi$ and a U-Net denoiser $f_\theta$ operating over a continuous noise scale $\sigma$ (Karras et al.). Given a conditioning image $x$, we compute a latent $z_0 = \phi(x)$. To form a video latent tensor, we replicate $z_0$ across $T$ frames:

$$Z_0 = [z_0, \dots, z_0] \in \mathbb{R}^{T \times C \times H' \times W'}.$$

Let $\sigma_0$ denote the initial noise level from the SVD schedule. We perform one explicit Euler integration step over the ODE at $\sigma_0$ (classifier-free guidance disabled):

$$Z_1 = Z_0 + \Delta\sigma \, f_\theta(Z_0, \sigma_0, c), \tag{2}$$

where $c$ denotes SVD conditioning (e.g., time/frame embeddings, text/image prompts), and $\Delta\sigma$ is the step size.

We do not render frames; instead, we extract a U-Net hidden activation at the lowest spatial resolution on the downsampling path before the mid-block:

$$H \in \mathbb{R}^{T \times H_d \times W_d \times C_h} = \mathrm{Hidden}^{\mathrm{pre\text{-}mid}}(f_\theta, Z_1). \tag{3}$$

**Multi-image extension.** For multiple images $\{x_k\}_{k=1}^K$, we first compute their latents $\{z_0^{(k)}\}$. These are inserted as keyframes within $T$ frames at indices $i_k = \mathrm{round}(\mathrm{linspace}(0, T-1, K))$. We initialize $Z_0$ with copies of $z_0^{(1)}$ and set $(Z_0)_{i_k} \leftarrow z_0^{(k)}$ before the Euler step, yielding multi-image-aware $H$.

**Static+dynamics token fusion.** We convert $H$ into a token sequence by flattening spatial locations: $L = H_d W_d$, $\tilde{H} \in \mathbb{R}^{(T \cdot L) \times C_h}$. A projector $P_{\mathrm{svd}} : \mathbb{R}^{C_h} \to \mathbb{R}^d$ maps these to the LLM token space:

$$V_d = P_{\mathrm{svd}}(\tilde{H}) = \mathrm{MLP}_d(\tilde{H}) \in \mathbb{R}^{M \times d}, \tag{4}$$

where $M = T \cdot L$.

The SigLIP tokens $\hat{V}_s$ (Eq. 1) are concatenated with $\hat{V}_d$ to form the visual sequence:

$$V = [\hat{V}_s; \hat{V}_d].$$

Table 4: Model Performance Across Different Frame Numbers. These are DyVA with SVD only encoders using 5761024

| Frames | Pretrain | Tuning | VQAv2 | GQA | VizWiz | VSR (BL:51) | POPE | TallyQA | SeedBench | SpatialMM-Obj | 3DSR-Bench-real |
|---|---|---|---|---|---|---|---|---|---|---|---|
| 1 | 558k | 665k | 59.38 | 47.75 | 48.74 | 52.12 | 75.74 | 50.97 | 51.12 | 38.81 | 45.40 |
| 4 | 558k | 665k | 60.10 | 47.36 | 46.24 | 53.19 | 77.60 | 50.68 | 52.24 | 42.48 | 45.67 |
| 8 | 558k | 665k | 60.80 | 48.63 | 50.25 | 52.20 | 78.15 | 51.46 | 52.81 | 37.98 | 46.32 |
| 14 | 558k | 665k | 61.73 | 49.71 | 38.68 | 53.43 | 78.80 | 52.19 | 53.28 | 39.78 | 46.32 |

Table 5: **SVD vs. SVD-MiddleBlock.** Comparison of different fusion strategies using SVD latents.

| Model | VQAv2 | GQA | VizWiz | VSR | POPE | TallyQA | SeedBench | Spatial | 3DSR |
|---|---|---|---|---|---|---|---|---|---|
| **DyVA-SVD** | 61.82 | 50.20 | 50.60 | 53.60 | 75.61 | 53.27 | 52.55 | 40.60 | 43.50 |
| **DyVA-SVD-Post-MiddleBlock** | 62.86 | 54.30 | 51.41 | 57.69 | 80.17 | 51.36 | 52.50 | 41.13 | 43.84 |

## D.1 U-NET LAYER CHOICE

**Passing through deeper layers in the UNet allows models to obtain better results.** Extracting latents from deeper U-Net blocks (pre-mid vs. mid) changes the balance between global layout information and fine-grained motion cues. Better latents seem to We leave exploration of the utilization of different latents from the world model as future work.

