# OpenReview forum: "Can World Models Benefit VLMs for World Dynamics?"
_ICLR.cc/2026/Conference — ICLR 2026 Conference Withdrawn Submission_

### Official Review · Reviewer_vh3z · 2025-10-31

**Soundness:** 3
**Presentation:** 3
**Contribution:** 2
**Rating:** 2
**Confidence:** 4

**Summary:**

This paper introduces DyVA, a framework for enhancing Vision-Language Models (VLMs) by integrating generative priors from a pre-trained video diffusion model (SVD). The core idea is to fuse standard static visual features (from SigLIP) with dynamic, predictive latents extracted from a single denoising step of SVD. The authors claim this "World-Language Model" (WorldLM) paradigm significantly boosts spatial and multi-frame reasoning, even when the VLM alignment stage is trained exclusively on single images. The model achieves strong performance on several benchmarks, most notably MindCube.

**Strengths:**

- Recent studies have reported favorable results demonstrating the efficacy of extracting predictive features from pretrained generative models for other models (e.g., https://arxiv.org/abs/2411.09153, https://arxiv.org/abs/2412.14803). This work, therefore, aligns well with and contributes to this emerging trend in feature representation learning.
- The model achieved state-of-the-art performance on the complex MindCube benchmark (which requires 3D mental rotation and perspective-taking) when compared against its reported baselines, including GPT-4o and Qwen-2.5-VL.
- It seems to have the effect of augmenting information as if there were multi-frames even in a single-image setting, and I think this is a practical direction.

**Weaknesses:**

While the paper presents a promising direction with strong empirical results, the submission suffers from methodological and framing issues, in particular around terminology, credit attribution, missing baselines, and ablations that risk misleading readers about what is
actually being demonstrated.

- The diffusion feature's properties and performances of downstream tasks using it are critically dependent on the chosen denoising timestep (https://arxiv.org/abs/2303.16203, https://arxiv.org/abs/2412.03439). The paper fails to analyze this crucial hyperparameter, simply stating a "single denoising step" is used.
- In common usage, a world model implies controllability (e.g., action- or goal-conditioned rollouts). Here the SVD prior is image-conditioned but action-agnostic and used for a single denoising step, so it functions as a generative video prior rather than a controllable simulator. I recommend the authors clarify terminology or temper the ‘world model’ claim.
- The "world model" features, which should intuitively excel at predicting temporal dynamics, but the model's strengths are confined to static 3D spatial reasoning (orientation, layout). This suggests the SVD is not acting as a “world model” but merely as a powerful, implicitly-trained 3D feature extractor. I think that this issue stems from acquiring features without conditioning prompts. Could the authors demonstrate or visualize what semantic content the SVD features used in VLM training actually encapsulate?
- Also, given this context, this study should include a comparison with approaches that utilize pretrained 3D features (e.g., https://arxiv.org/abs/2503.11651) as their vision features for VLMs. I think VLMs can be trained in a single-image training scheme
with pretrained 3D vision encoders.
- The ‘emergent’ phrasing is misleading: the multi-frame skill is transferred from the video-pretrained SVD prior, not learned from single-image data. The result is valuable practically (video gains without video-supervised alignment), but the claim should attribute the source of competence.
- The paper's central thesis is that the “generative” nature of the encoder of video diffusion is key. To prove this, the authors must compare DyVA against features from strong non-generative video models (e.g.,https://arxiv.org/abs/2203.12602,
https://arxiv.org/abs/2506.09985) and generative image models (https://arxiv.org/abs/2112.10752). The current comparison against only VAE-Only is insufficient, as it merely proves that a temporal model is better than a static one, not that a “generative video model” is superior.
- The claims regarding DyVA's benefits on models like Qwen2.5 (e.g., on TallyQA) are inconclusive. The authors fail to provide the most critical baseline: the performance of the base Qwen2.5-VL model without DyVA features on this specific benchmark. Without this comparison, it is impossible to attribute the reported gains to the proposed method rather than the inherent strength of the Qwen2.5 backbone itself.

**Questions:**

- The paper argues that SigLIP is crucial because it provides "text-aligned" features, which SVD-Only lacks. But I think SVD is also trained with text caption and video and is therefore text-aligned. Can the authors further elaborate on the specific distinction between the text alignment provided by SigLIP and the alignment naturally present in SVD features? Also, can the authors show results using text prompts in SVD-only setting?
- While a concurrent study evaluated fifteen video-supporting MLLMs (https://arxiv.org/abs/2412.14171), the evaluation criteria or experimental settings used were considered not sufficiently comprehensive to draw definitive conclusions about their performance.

---

### Official Review · Reviewer_3qso · 2025-10-31

**Soundness:** 2
**Presentation:** 2
**Contribution:** 2
**Rating:** 2
**Confidence:** 4

**Summary:**

This paper proposes World-Language Models (WorldLMs), which incorporates pre-trained world models into the multimodal large language model (MLLM) to enable multi-frame reasoning. The study systematically compares paradigms, explores design choices, and diagnoses benchmark performance, suggesting gains in spatial and temporal reasoning despite semantic limitations.

**Strengths:**

1. The concept of repurposing video generation models as encoders for VLMs is well-motivated and shows potential on enhancing VLMs' multi-frame reasoning capabilities.
2. The paper provides extensive analyses of frame budgets (1-14), fusion strategies (early/late), and encoder layers, which are thorough and reveal key trade-offs.

**Weaknesses:**

1. Prior works have explored enhancing VLMs with multi-frame reasoning capabilities, such as Video LLM (Yuan et al., 2025) or Dynamic-VLM (Wang et al., 2025). However, the paper lacks discussions on the relationships between the proposed method and these works, as well as direct comparisons with these closely related approaches.
2. Experimental details are insufficient. The configurations for DyVA-7B and DyVA-Qwen2.5-7B are not clearly specified, and the baselines for these models are unclear, leading to confusion about the results. Moreover, to highlight the contributions of the world model, an ablation study comparing performance with and without the generative encoder is essential.
3. The improvements from the proposed method appear marginal. For example, it offers only trivial gains over single-frame baselines (e.g., 55.24 vs. 53.16 for DyVA-Qwen2.5-7B and Qwen2.5-VL-7B on SAT Synthetic, as shown in Table 1). While the authors attribute this to degraded semantic capabilities, the integration of a powerful semantic encoder like SigLIP raises questions about how to achieve more substantial enhancements.
4. Some claims are inappropriate or unsubstantiated.
   - The authors claim that the model achieves emergent multi-frame reasoning without multi-image pre-training. However, the generative encoder is pretrained on video generation, which inherently involves multi-frame data, making the "zero-shot" designation potentially misleading.
   - Furthermore, the core question addressed is the title's Q1: "Can World Models Benefit VLMs for World Dynamics?" Yet, the introduction poses Q2: "To truly understand the world, must a model first learn to predict?" The paper does not resolve Q2, as it only demonstrates benefits from world models without examining the necessity of predictive learning for true understanding.

[R1] Y. Yuan, H. Zhang, W. Li, et al. "Videorefer Suite: Advancing Spatial-Temporal Object Understanding with Video LLM," in Proc. IEEE/CVF Int. Conf. Comput. Vis. Pattern Recognit. (CVPR), 2025.

[R2] H. Wang, Y. Nie, Y. Ye, et al. "Dynamic-VLM: Simple Dynamic Visual Token Compression for VideoLLM," in Proc. IEEE/CVF Int. Conf. Comput. Vis. (ICCV), 2025.

**Questions:**

1. **Related Work**: Could you discuss how WorldLM differs from Video LLM (Yuan et al., 2025) and Dynamic-VLM (Wang et al., 2025) in architecture, training, and use of video priors? Adding direct comparisons would strengthen the contribution.

2. **Ablations & Details**: Could you clarify the exact configurations of DyVA-7B and DyVA-Qwen2.5-7B (e.g., projectors, fusion, hyperparams) and specify baseline sources? An ablation comparing the model with and without the generative encoder (from SVD) would clearly isolate the generative encoder’s impact.

3. **Improvement Magnitude**: Given modest gains (e.g., ~2% on SAT Synthetic), what are the main bottlenecks despite using strong SigLIP? Any preliminary results on amplifying dynamic priors (e.g., fusion tuning, higher resolution)?

---

### Official Review · Reviewer_8MoX · 2025-10-31

**Soundness:** 3
**Presentation:** 2
**Contribution:** 3
**Rating:** 4
**Confidence:** 4

**Summary:**

This paper investigates how the priors learned by video diffusion models can improve vision-language understanding. They repurpose Stable Video Diffusion (SVD) as a vision encoder, and align it with MLLMs. Experimental results show that SVD, together with SigLIP achieves strong results on 3D-related, OOD MLLM understanding tasks, outperforming SOTA models such as GPT 4o.

**Strengths:**

- This paper is clearly written and easy to follow.
- DyVA achieves strong OOD results compared with SOTA models.
- Aligning video generation models with MLLM topic is very interesting.

**Weaknesses:**

One major weakness is the setting of the evaluation:
- Only a single SVD backbone is used, and only the variant of SigLIP + SVD brings strong improvements.
- How would the model work with other generative encoders -- both image-based ones like Stable Diffusion, SD3, and DiT-based video diffusion models like CogVideoX, Wan, etc.
- Why is the OOD evaluation the main focus? It does not hold from the training data side -- If the paper reuses LLaVA's training data, all the evaluation datasets from LLaVA are OOD evaluation. If the perception, 3D, or video is the main focus, then a lot of datasets, such as the ones used by Cambrian, are also strong candidates. The current major evaluation benchmarks are not widely used standard benchmarks, making it doubtful for the effectiveness of the method.
- Following the previous comment, the major evaluation benchmarks are OOD for MLLMs, but they are not necessarily OOD for world models, making the variant with an SVD encoder naturally better performing on these benchmarks.


Some other minor things are:
- Limited supervision story: Attempts to adapt U‑Net/VAE with text loss failed; alternative alignment routes are only hypothesized but not demonstrated.
- The “envisioning” prior sometimes harms fine‑grained recognition; more quantification of hallucinations (beyond a case study) would strengthen claims

**Questions:**

- What is the training data for DyVA? Based on Table 2, is it the SFT data from llava 1.5?
- How about training and inference cost, time, and memory cost, comparison compared with the Siglip-only baseline?
- For Table 3, it is better to add a reference number for the default model to check how the SVD encoder affects general VQA tasks. Taking Llava-1.5-7B as an anchor point, it seems that for a lot of tasks, even the best model still lagged behind.
- How would other world models, like VJepa, work with MLLMs?

---

### Official Review · Reviewer_5Wj5 · 2025-11-01

**Soundness:** 2
**Presentation:** 1
**Contribution:** 2
**Rating:** 2
**Confidence:** 3

**Summary:**

The authors present a simple approach of adding video diffusion features as input to LLMs to solve VQA and investigate how useful these generative features are for single and multi image reasoning. The authors evaluate their method zero-shot on several "OOD" benchmarks and run several ablations with existing "semantic" encoders like SigLIP and DINO.

**Strengths:**

1. The authors achieve SOTA performance on the MindCube dataset.
2. The authors show that their method can generalise to multi-image reasoning even though it is fine-tuned in a single image setting.
3. The authors evaluate their method on several benchmarks both multi-image and single-image.
4. The authors perform ablations with different encoders like SigLIP and DINO.

**Weaknesses:**

1. The results are very poorly reported. In Table 2, there is no mention of the Qwen only model's results, so it is not possible to judge the effect of adding the diffusion features.
2. In table 1, for the MMSI and SAT synthetic the improvements upon adding the video diffusion features to Qwen is marginal. The authors do not investigate the disparity between these results and MindCube where the improvement is quite large.
3. The authors claim that their evaluation benchmarks are Out-of-Domain, but do not mention how so. Just because the model was not trained on this dataset does not make it OOD.
4. The authors do not adequately demonstrate/back their claim of a "shift in reasoning paradigm".


Overall it seems like the authors present a very simple approach of using video diffusion model features as input to LLMs to solve VQA tasks and it gives mostly marginal improvements on different benchmarks except Mindcube.

**Questions:**

see weaknesses

---

### Note · Authors · 2025-12-08

I have read and agree with the venue's withdrawal policy on behalf of myself and my co-authors.